# Dynamics of HIV Reservoir and HIV-1 Viral Splicing in HCV-Exposed Individuals after Elimination with DAAs or Spontaneous Clearance

**DOI:** 10.3390/jcm11133579

**Published:** 2022-06-21

**Authors:** Paula Martínez-Román, Celia Crespo-Bermejo, Daniel Valle-Millares, Violeta Lara-Aguilar, Sonia Arca-Lafuente, Luz Martín-Carbonero, Pablo Ryan, Ignacio de los Santos, María Rosa López-Huertas, Claudia Palladino, María Muñoz-Muñoz, Amanda Fernández-Rodríguez, Mayte Coiras, Verónica Briz

**Affiliations:** 1Laboratory of Reference and Research on Viral Hepatitis, Centro Nacional de Microbiología, Instituto de Salud Carlos III, 28220 Madrid, Spain; paula.mroman@isciii.es (P.M.-R.); celicres@externos.isciii.es (C.C.-B.); da.valle@isciii.es (D.V.-M.); violeta.lara@isciii.es (V.L.-A.); soniarca@ucm.es (S.A.-L.); 2Instituto de Investigación Sanitaria Hospital de la Paz (IdiPAZ), 28046 Madrid, Spain; lmcarbonero@gmail.com; 3Department of Infectious Diseases, Infanta Leonor Hospital, 28031 Madrid, Spain; pabloryan@gmail.com; 4Servicio de Medicina Interna-Infecciosas, Hospital Universitario de La Princesa, 28006 Madrid, Spain; isantosg@hotmail.com; 5Immunopathology Unit, Centro Nacional de Microbiología, Instituto de Salud Carlos III, 28220 Madrid, Spain; mariarosa.lopezhuertas@gmail.com (M.R.L.-H.); mcoiras@isciii.es (M.C.); 6Faculty of Pharmacy, Research Institute for Medicines (iMed.ULisboa), Universidade de Lisboa, 1649-003 Lisbon, Portugal; cpalladino@edu.ulisboa.pt; 7Department of Animal Genetics, Instituto Nacional de Investigación y Tecnnología Agraria y Alimentaria (INIA), 28040 Madrid, Spain; mariamm@inia.csic.es

**Keywords:** HIV/HCV, HIV reservoir, viral splicing, dynamics, coinfection, DAAs, spontaneous HCV clearance

## Abstract

Background: Although human immunodeficiency virus type 1 (HIV-1) reservoir size is very stable under antiretroviral therapy (ART), individuals exposed to the Hepatitis C virus (HCV) (chronically coinfected and spontaneous clarifiers) show an increase in HIV reservoir size and in spliced viral RNA, which could indicate that the viral protein regulator Tat is being more actively synthesized and, thus, could lead to a higher yield of new HIV. However, it is still unknown whether the effect of HCV elimination with direct-acting antivirals (DAAs) could modify the HIV reservoir and splicing. Methods: This longitudinal study (48 weeks’ follow-up after sustained virological response) involves 22 HIV+-monoinfected individuals, 17 HIV+/HCV- spontaneous clarifiers, and 24 HIV+/HCV+ chronically infected subjects who eliminated HCV with DAAs (all of them aviremic, viral load < 50). Viral-spliced RNA transcripts and proviral DNA copies were quantified by qPCR. Paired samples were analyzed using a mixed generalized linear model. Results: A decrease in HIV proviral DNA was observed in HIV+/HCV- subjects, but no significant differences were found for the other study groups. An increased production of multiple spliced transcripts was found in HIV+ and HIV+/HCV+ individuals. Conclusions: We conclude that elimination of HCV by DAAs was unable to revert the consequences derived from chronic HCV infection for the reservoir size and viral splicing, which could indicate an increased risk of rapid HIV-reservoir reactivation. Moreover, spontaneous clarifiers showed a significant decrease in the HIV reservoir, likely due to an enhanced immune response in these individuals.

## 1. Introduction

Human immunodeficiency virus type 1 (HIV-1) infection affects 38 million people globally, with 1.7 million new cases in 2019 [1]. Around 2.75 million also live with Hepatitis C virus (HCV), since both viruses share transmission routes [2]. HIV/HCV coinfection alters the natural course of both diseases, worsening the condition of subjects who suffer from lower spontaneous HCV clearance rates, increased immune exhaustion, accelerated liver fibrosis, and incremented inflammatory responses [3,4,5,6].

Since 2013, the successive approval of new direct-acting antivirals (DAAs), most of them IFN-free regimens, has led to the curing of HCV infection in more than 90% of individuals [7]. However, the effect of DAAs on the immune system is still unknown. By contrast, HIV infection cannot be eradicated due to the very early establishment of viral reservoirs, to which current antiretroviral therapy (ART) is inaccessible [8].

Resting memory (r)CD4+ T-cells comprise most of the HIV viral reservoir that includes central-memory stem cells, central-memory T cells, effector-memory T cells, and transitional-memory T-follicular-helper cells. The latent cellular reservoir is established when previously infected active cells return to a resting state, and replication is hindered [9,10,11]. Nonetheless, the HIV viral reservoir includes other peripheral blood mononuclear cells (PBMCs), such as regulatory T cells [12], monocytes [13], NK cells, and γδT cells [14]. HIV reservoir size is very variable among people living with HIV (PLWH), even on ART, depending on different factors. However, although it might be stable in size after long-term ART, it is certainly dynamic in composition [15]. In fact, even when virological control is achieved, low-level viral replication continues in infected cells [16].

Viral splicing is necessary to create mature mRNA molecules and balance the expression of the different viral genes. Structural proteins Gag and Pol are coded in unspliced mRNA transcripts; Vif, Vpu, Vpr, and Env are coded in single-spliced transcripts; and regulatory proteins Rev, Nef, and Tat are coded in multiple-spliced transcripts [17]. Tat expression is indispensable for HIV transcription and elongation as well as for increased viral splicing. High quantities of inducible multiple-spliced transcripts could, thus, result in high concentrations of Tat protein, which result in an increased HIV productive infection [18].

Previously, our group described how HIV/HCV coinfection increases HIV reservoir size in rCD4+ T cells from HCV spontaneous clarifiers (HIV+/HCV-) and HCV chronic-infected individuals (HIV+/HCV+), in comparison with HIV-monoinfected individuals [19]. We also disclosed that HIV+/HCV+-coinfected subjects showed an increase in multiple-spliced transcripts compared to HIV-monoinfected PLWH [20]. Both findings suggest that the elimination of the HIV viral reservoir in HCV-exposed subjects may be hampered.

In this study, we analyzed the dynamics of HIV reservoir size and splicing in rCD4+ T cells, the main HIV reservoir, and rCD4+ T cells-depleted PBMCs (rCD4 T- PBMCs) in HCV-exposed individuals, including both HIV+/HCV- spontaneous clarifiers and HIV+/HCV+ chronically infected individuals who eliminated HCV infection with DAA administration, with the aim of analyzing the effect of HCV clearance (either spontaneous or DAA-mediated) on the HIV reservoir.

## 2. Methods

### 2.1. Patients

This is a prospective longitudinal study of 63 Caucasian HIV-1 infected adults under suppressive ART, recruited from three tertiary hospitals in the Comunidad de Madrid (Spain), belonging to the COVIHEP network (Supplementary File S1). All patients had been undetectable for HIV during the previous year and had CD4+ T-cells ≥ 500 cel/mm^3^, since at least one year before sample collection. No blips were observed during the study period. Exclusion criteria were hepatic decompensation, alcohol-related liver damage, presence of viral hepatitis B antigens or antibodies against viral hepatitis B, opportunistic infections, drug abuse and addiction, other diseases (diabetes, nephropathies, autoimmune diseases, hemochromatosis, cryoglobulinemia, primary biliary cirrhosis, Wilson’s disease, deficiency of alpha1 antitrypsin, and neoplasia), and pregnancy. Appendix A includes the main characteristics related to the study design.

In HIV+/HCV+ individuals, baseline samples were obtained just before DAA-treatment initiation, and endpoint samples were obtained 48 weeks after sustained virological response. At the baseline, none of these patients had ever taken any treatment for HCV infection. However, all persons living chronically with HIV and HCV successfully cleared HCV infection with DAAs and obtained a sustained virological response (SVR) at the endpoint. In HIV+ and HIV+/HCV- individuals, the baseline and endpoint samples also differed after 48 weeks. The epidemiological and clinical variables of the study patients were collected from medical records.

Three groups of PLWH were included: (1) HIV+ control group (*n* = 22): PLWH who were negative for both HCV PCR and antibodies; (2) HIV+/HCV- group (*n* = 17): PLWH who had been infected with HCV and had spontaneously cleared HCV infection during the first 6 months after HCV infection (negative HCV PCR but positive HCV antibodies); and (3) HIV/HCV+ group (*n* = 24): PLWH chronically infected HCV (positive HCV PCR and antibodies) naive to any HCV treatment at the baseline, achieving SVR with DAAs at the endpoint. The STROME-ID checklist was used to strength study design [21].

### 2.2. Purification of PBMCs and DNA/RNA Extraction

PBMCs were obtained by means of a density-gradient centrifugation from 50 mL of blood and rCD4+ T-cells (CD4+CD25−CD69−HLA/DR−) that were isolated by negative selection using an EasyStep Human Resting CD4+ T Cell Isolation kit (StemCell Technologies, Vancouver, Canada). The purity of the CD4+CD25-HLA/DR-CD69- T cells was assessed by flow cytometry as >99%. Total DNA/RNA from both rCD4+ cells and rCD4 T- PBMCs were extracted using an AllPrep^®^ DNA/RNA Mini Kit (Qiagen, Hilden, Germany).

### 2.3. Quantification of HIV-Reservoir Size

Nested Alu-HIV-LTR PCR was used to measure integrated HIV DNA in rCD4+ T cells and rCD4 T- PBMCs, as previously described [19,22]. HIV-reservoir-size results were expressed as the number of HIV viral DNA copies integrated per 10^6^ cells, given as the arithmetic mean and standard error of mean (SEM) for each group. qPCR reactions were carried out with StepOne software v2.3 (Applied Biosystems, Waltham, MA, USA).

### 2.4. HIV Viral-Splicing Analysis

Viral-splicing forms were quantified in rCD4+ T cells and rCD4 T- PBMCs by q-RT-PCR, as described previously [20,23]. Briefly, 250 ng of extracted total RNA were used in a master mix containing 5× buffer, 25 mM ClMg, dNTPs, RNAsin, and retrotranscriptase. Samples were then incubated at 25 °C for 5 min, 45 °C for 1 h, and 70 °C for 15 min in a C1000 thermal cycler (BioRad, Hercules, CA, USA). Four different qPCRs were carried out, in order to measure the different forms of viral splicing, using GAPDH as a housekeeping gene. qPCR was performed by adding 2 μL of cDNA to Taqman Universal PCR Master Mix 2× (Applied Biosystems, Waltham, MA, USA) and TaqMan probes FAM/Zen/Iowa Black (Integrated DNA Technologies, Leuven, Belgium), then primers for each specific target were used, as described before with modifications [23]. Primers and Taqman probes used for each target were as follows: unspliced (mf299 primer, mf302 primer, and mf348 TaqMan probe); single spliced (mf222 primer, mf83 primer, and mf226m TaqMan probe modified to 5′-ACCCGACAGGCCCGAAGGAA-3′); multiple spliced (mf84-AK145 primer, mf83 primer, and mf226m TaqMan probe); and GAPDH (GAPDH_forward primer, GAPDH_reverse primer, and GAPDH TaqMan probe). PBMCs infected in vitro with NL4-3 HIV strain were used as a positive control for active infection (0.2 ng p24/μL). qPCRs were carried out in a StepOne thermal cycler (Applied Biosystems) using a 96-well plate. Amplification results for the different forms of viral splicing were expressed as a fold-change (ΔRQ) with respect to the amplification of the active-infection positive control. RT-q-PCR reactions were carried out with StepOne software v2.3. (Applied Biosystems).

### 2.5. Statistical Analysis

Epidemiological and clinical characteristics of the study population were presented as the median and interquartile range (IQR) for continuous variables as well as a percentage for categorical variables. Differences between independent groups were evaluated with the χ^2^ test for categorical variables, and with an ANOVA test and a Kruskal–Wallis H test after studying normalization, with a Shapiro–Wilk test for continuous variables. The Wilcoxon test was used for continuous variables to compare paired samples between the baseline and endpoint of the study.

Gamma-distributed univariate and multivariate generalized-linear models (GLM) with a long-link function were used to evaluate the differences in reservoir size between the HIV+ control group and the HIV+/HCV- and HIV+/HCV+ groups. This model was adjusted by the time of HIV infection, ART, and total CD4+ T cells. Additionally, a mixed GLM with gamma distribution was used to evaluate repeated measurements of HIV reservoir size and viral splicing between time points (baseline or endpoint).

These tests give the value of the arithmetic mean ratio (AMR). ΔRQ means and SEM of viral splicing were calculated for the three forms of HIV splicing.

All statistical analyses were executed using SPSS v 22.0 (SPSS Inc., Chicago, IL, USA) and R statistical software (v4.0.2), (Vienna, Austria). *p* < 0.05 (2-tailed) was defined as statistical significance.

## 3. Results

### Study Subjects

A total of 63 Caucasian participants were classified according to their HCV status: (1) HIV+ group: 22 PLWH; (2) HIV+/HCV- group: 17 PLWH previously exposed to HCV who spontaneously cleared HCV infection; and (3) HIV+/HCV+ group: 24 PLWH also chronically coinfected with HCV, who had never received any treatment for HCV at the baseline but eliminated the hepatitis virus at endpoint with DAAs.

Table 1 summarizes the main epidemiological and clinical differences between the study patients. Individuals were Caucasian and 54.0% (*n* = 34) were men. The median age was 52.0 years (IQR: 44.0–55.0), and the median time of HIV infection was 20 years (10.5–26.5). Differences between study groups were found in terms of HIV-transmission route, with sexual transmission being predominant in HIV+ group (*n* = 16, 72.8%) and the parenteral route in HCV-exposed subjects (*n* = 21, 51.2%) (*p* < 0.001). Differences were also found regarding the IFNL3 genotype, as 15 (88.2%) spontaneous clarifiers presented the CC favorable genotype (*p* < 0.001). Although almost one-in-four individuals acquired AIDS (*n* = 14, 22.2%), no differences between groups were found in the Center for Disease Control and Prevention classification system for HIV infection (*p* = 0.325). There were also no differences in relation to ART regimens, with most PLWH receiving an integrase inhibitor-based regimen (INIs) (*n* = 27, 42.9%) (*p* = 0.409). When stratifying patients on an INSTI vs. a no INSTI regimen, no differences were observed in HIV reservoir or HIV viral splicing.

## 4. Patients

HIV+ group: PLWH, who never were exposed to HCV (negative for both HCV PCR and antibodies).HIV+/HCV- group: PLWH coinfected with HCV, who spontaneously cleared HCV infection (negative HCV PCR but positive for HCV antibodies).HIV+/HCV+ group: PLWH chronically infected HCV (positive for both HCV PCR and antibodies), who had never been treated for HCV at the baseline but who achieved SVR with DAAs at the endpoint.

Table 2 shows the lymphocytic subpopulations of the study population. For all study participants, immune status showed no differences between the baseline and endpoint in relation to the total number of CD4+ T-cells ((783.0 cells/µL, 572.0–1099.0) vs. (835.0 cells/µL, 609.0–1117.0)) (*p* = 0.939) and total CD8+ T cells ((917.5 cells/µL, 803.3–1492.6) vs. (962.0 cells/µL, 794.3–1427.8)) (*p* = 0.784). There were also no differences when comparing CD4+/CD8+ ratios between the baseline (0.93, 0.62–1.18) and endpoint (0.96, 0.74–1.18) (*p* = 0.782).

Table 3 describes the clinical characteristics of HCV infection. The majority of HIV+/HCV+ individuals were infected with HCV genotypes 1 (*n* = 12, 50.0%) and 4 (*n* = 8, 33.3%) and were receiving Ledipasvir + Sofosbuvir as treatment to eliminate HCV infection (*n* = 15, 62.5%). No differences were found in relation to the fibrosis stage, as most HCV-exposed individuals (*n* = 39, 61.9%) were diagnosed with the F0–F1 fibrosis stage (*p* = 0.066).

### HIV Reservoir Size and Dynamics

Changes in HIV-reservoir size for paired samples of rCD4+ T cells were analyzed in the three groups of PLWH. The HIV+/HCV+ group showed an increase in the viral-reservoir size at the end of the study, although this increase was not statistically significant (mean: 680.29 ± 207.82 copies vs. 2398.81 ± 1572.16 copies; *p* = 0.248) (Figure 1a). HIV-monoinfected individuals showed a similar tendency (HIV+: 279.54 ± 69.00 copies at the baseline vs. 956.51 ± 74.00 copies at endpoint; *p* = 0.180). By contrast, the analysis revealed a statistically significant decrease in HIV-reservoir size in HIV+/HCV- spontaneous clarifiers (1276.22 ± 419.46 copies vs. 852.06 ± 232.06 copies; *p* = 0.048) between the baseline and final endpoint.

The dynamics of the viral-reservoir size were also determined in resting CD4 T cells-depleted PBMCs (rCD4 T- PBMCs). In contrast to what was observed in rCD4 T cells, none of the study groups showed significant differences in HIV-reservoir size in these cells ((HIV+(mean): 249.45 ± 90.85 copies vs. 607.84 ± 251.47 copies; *p* = 0.060) (HIV+/HCV-: 599.04 ± 268.34 copies vs. 966.23 ± 445.95 copies; *p* = 0.104) (HIV+/HCV+: 986.32 ± 866.56 copies vs. 457.48 ± 252.33 copies; *p* = 0.777)) (Figure 1b).

Appendix A show cross-sectional studies of HIV-reservoir size (rCD4+ T cells and rCD4 T- PBMCs, respectively) and the resulting AMR values and statistical significance for the comparison of the HCV-exposed groups (HIV+/HCV- and HIV+/HCV+) with the HIV-monoinfected group at the baseline and endpoint, by means of univariate and multivariate GLM.

Appendix A shows patients’ pro-viral DNA copies at the baseline vs. the endpoint in rCD4+ T cells and rCD4 T-PBMCs.

Changes in the expression of HIV viral-splicing transcripts in rCD4+ T cells were determined for the paired samples from the three study groups. The analysis of unspliced transcripts showed a significant increase in expression (*p* < 0.001) in all study groups (HIV+ group: mean 78.38 ± 23.99 ΔRQ at baseline vs. 19,642.97 ± 12,755.41 ΔRQ at endpoint; HIV+/HCV- group: 243.19 ± 98.75 ΔRQ vs. 25,931.78 ± 9188.52 ΔRQ; HIV+/HCV+ group: mean 67.93 ± 22.47 ΔRQ vs. 6720.26 ± 1730.39 ΔRQ) (Figure 2a). The analysis of single-spliced transcripts also showed a significant increase in the mean expression in all study groups (*p* < 0.001) (HIV+: 61.84 ± 57.51 ΔRQ vs. 6017.93 ± 2263.02 ΔRQ; HIV+/HCV-: 60.49 ± 31.17 ΔRQ vs. 2937.77 ± 1164.09 ΔRQ; HIV+/HCV+: 390.25 ± 285.76 ΔRQ vs. 915.50 ± 508.29 ΔRQ). However, in the analysis performed in multiple-spliced transcripts, only coinfected subjects showed a significant increase in the mean expression (*p* < 0.001) (0.05 ± 0.04 ΔRQ vs. 43.80 ± 39.3 ΔRQ), in contrast with the spontaneous clarifiers (0 ± 0 ΔRQ vs. 159.02 ± 58.98 ΔRQ; (*p* = 0.262)) and HIV-monoinfected individuals (3.06 ± 0.02 ΔRQ vs. 45.29 ± 3.01 ΔRQ; (*p* = 0.292)).

The dynamics of the expression of HIV viral-splicing transcripts were also analyzed in rCD4 T- PBMCs (Figure 2b). As occurred with rCD4+ T cells, unspliced transcripts showed a significant increase in expression for all study groups (*p* < 0.001) (HIV+: 30.92 ± 12.20 ΔRQ vs. 17,138.64 ± 12,087.15 ΔRQ; HIV+/HCV-: 1020.66 ± 368.41 ΔRQ vs. 56,927.92 ± 23,532.99 ΔRQ; HIV+/HCV+: 2930.97 ± 2602.19 ΔRQ vs. 80,486.61 ± 73,504.39 ΔRQ). Nevertheless, the analysis did not show any significant increase for single- or multiple-spliced transcripts. Single-spliced mean values were as follows: HIV+ group: 64.87 ± 47.21 ΔRQ vs. 2792.83 ± 1869.06 ΔRQ (*p* = 0.308); HIV+/HCV- group: 1282.71 ± 838.93 ΔRQ vs. 340,228.94 ± 312,833.13 ΔRQ (*p* = 0.096); and HIV+/HCV+ group: 3020.31 ± 2540.54 ΔRQ vs. 29,973.98 ± 27,904.11 ΔRQ (*p* = 0.362). Multiple-spliced mean values were as follows: HIV+ group: 2.80 ± 0.80 ΔRQ vs. 509.98 ± 9.98 ΔRQ (*p* = 0.287); HIV+/HCV- group: 0 ± 0 ΔRQ vs. 6.80 ± 6.80 ΔRQ (*p* = 0.304); and HIV+/HCV+ group: 0.03 ± 0.01 ΔRQ vs. 518.14 ± 12.64 (*p* = 0.371)

Appendix A show the cross-sectional expression means and statistical significance for the comparison of chronically coinfected individuals and spontaneous clarifiers with monoinfected individuals, by means of a Kruskal–Wallis H-test.

## 5. Discussion

Understanding the effect of HCV infection on the size of the HIV reservoir is important for improved clinical management of HIV/HCV-coinfected individuals. HIV+/HCV+ subjects, who were never treated for HCV infection, showed a significantly larger reservoir size than HIV-monoinfected individuals, as previously described by our group [24]. These differences in HIV-reservoir size were maintained after follow-up, following HCV elimination with DAAs. While it has been found that the HIV-1 reservoir remains stable after the administration of DAAs in coinfected subjects [24,25,26], this is the first study carried out in rCD4+ T cells, the main HIV reservoir. Our results contrast with those observed by Álvarez and colleagues [27] who did not observe differences between HIV-monoinfected and -coinfected individuals. Differences relating to patients’ inclusion and exclusion criteria, Fiebig stage, time since HIV diagnosis, cells in which proviral DNA was quantified, and the different methodology used could explain the different results. However, both studies present a similar number of patients, so further studies including a higher number of patients should be valuable.

The overall stability of the reservoir size may be due to successful ART, which avoids viral replication, suppresses viral load, and subsequently stabilizes the latent HIV reservoir [28], which is then preserved by continuous replenishment by the homeostatic cytokines [10]. This phenomenon could be stronger in coinfected individuals, due to constant immune activation, even after HCV elimination due to liver stiffness [29], epigenetic modifications [30], and exhaustion of the immune system [31]. In fact, it has been previously found that liver-virus elimination through DAAs fails to normalize the values of frequency and activation of the peripheral rCD4+ T cells [32], HCV-specific CD8+ T cell responses [31], and cytokine levels, such as IL-4, IL-17,IL-1b, IL-2, IFN-γ, and TNF-α [25,33,34]. The persistence of this altered lymphocytic and inflammatory state could favor reservoir maintenance through the clonal and homeostatic proliferation of infected cells.

An increase in the HIV reservoir was previously observed in HIV+/HCV- spontaneous clarifiers in comparison with HIV-monoinfected subjects [19]. Interestingly, HIV proviral DNA decreased significantly in rCD4+ T cells from the HIV+/HCV- group at the endpoint of the study, matching their reservoir size to that of the HIV+ group. Considering coinfection as a simultaneous phenomenon or as a repeated occurrence due to the continued use of parenteral drugs in some of these individuals, the larger HIV reservoir initially observed was likely due to a potent anti-HCV immune response in the spontaneous clarifiers. This immune activation is able to eliminate HCV infection but may also favor the persistence of the HIV reservoir, since parameters such as CD4+ T cell activation, NF-κβ signaling and IFN-γ are significantly elevated [35,36,37]. This immune response is normalized once HCV has cleared [35]. We did not observe a stabilization, however, but rather a decrease in the size of the HIV reservoir over time. Experimental strategies, intending to decrease or eradicate the HIV reservoir, include the activation of latently infected CD4+ T cells followed by subsequent elimination of the reactivated virus by increasing the killing rate of CD8+ T cells [38]. The lack of the literature on the effect of spontaneous HCV clearance on the HIV reservoir makes it difficult to discern a similar biological mechanism that could explain this reduction in our study group. Furthermore, it is worth mentioning that the expression of molecules such as HLA-B27, HLA-B57, HLA-Bw4–80, and KIR3DL1, which have been related to the spontaneous elimination of HCV [39,40,41], are also related to the control of HIV viral replication in elite controllers and long-term non-progressors [42,43].

There is a secondary reservoir in the fraction of PBMCs depleted from rCD4+ T cells that may also be established during the first days of infection and which remains stable over time even after ART [13,44]. We did not observe significant differences in the HIV reservoir, neither between study groups at the baseline or endpoint nor in the dynamics of its size. Since these cells do not comprise the main HIV reservoir, the immune activation caused by HCV exposure was not likely reflected in its establishment. To our knowledge, this is the first study to show the impact of HCV exposure on latent HIV infection in this type of cells.

Regarding HIV viral splicing, multiple-spliced mRNA transcripts are the best indicators of competent viral replication, since HIV splicing requires regulatory elements that are often defective in proviruses with large deletions [45,46]. In addition, they are essential for the production of the viral protein Tat, the main regulator for viral transcription and elongation [47,48]. Our data showed an increase, although not a significant one, related to multiple-spliced variants in HIV/HCV-coinfected individuals when compared to the HIV group. That could be in part because multiple-spliced transcripts are less abundant than unspliced RNAs and, therefore, are very difficult to detect in individuals on ART [45]. Further work including a higher number of subjects would be necessary in order to confirm differences in multiple-spliced variants in HIV/HCV-coinfected individuals. In addition, as expected, there was a higher overall viral transcription in rCD4 T- PBMCs (CD4+CD25+CD69+) than in rCD4+ T cells (CD4+CD25-CD69-), which indicated that HIV was mostly transcribed in activated cells that expressed essential transcription factors for HIV activation, such as NF-κB [49]. Moreover, in rCD4+ T cells, an increased expression of multiple-spliced transcripts was observed in the HIV+/HCV+ group. It is likely that persistent inflammation in these individuals leads to the homeostatic proliferation of latently infected cells, in which low-level viral replication occurs [10]. In fact, this is one of the possible reasons for the differences, previously observed by our group, between chronic HIV+/HCV+-coinfected subjects and monoinfected PLWH [25]. However, the overall stability of the HIV reservoir within this group of subjects indicates that this event does not result in a sufficient increase in the concentration of Tat to reactivate the latent provirus. A longer follow-up time would be necessary to study whether this increase in multiple-spliced mRNA stabilizes, as expected, in resting cells of PLWH under ART.

In the fraction of PBMCs depleted from rCD4+ T cells, we also observed an increase in the production of multiple-spliced mRNA in both HIV+ and HIV+/HCV+ individuals. Although this increase was not statistically significant, it is interesting to note that these subjects were not able to control viral replication despite ART [28]. Moreover, sustained immune activation despite HCV elimination in the HIV+/HCV+ group could favor viral replication, since the number of activated lymphocytes with integrated provirus is possibly increased [25,33,34]. The increase in multiple-spliced transcripts was much smaller in HIV+/HCV- than the other two study groups, within this fraction of cells. This proved that these subjects may control viral replication more effectively, and it is likely that those mechanisms that allow spontaneous clarification of HCV infection may also confer protection against other viral infections.

Finally, an increase in rCD4+ T cells and rCD4 T- PBMCs single-spliced and unspliced transcripts in rCD4+ T cells was observed over the follow-up period for all study groups. However, an increase in these transcripts may not be the best indicator of active viral infection. In fact, an increase in unspliced viral mRNA, with no increase in plasma RNA levels in HIV/HCV-coinfected subjects, following HCV clearance after administration of DAAs [25], could be due to an increased proportion of defective proviruses during ART. Although these proviruses are unable to produce infectious virions, they may initiate transcription and express viral proteins, which would be a source of persistent inflammation [50] that may support a continuous replenishment of the HIV reservoir.

This study presents some shortcomings that should be taken into consideration for the interpretation of the results. Study group 1 shows significantly shorter periods of HIV infection than the other groups, which might have impacted the findings in the lower DNA and the unspliced RNA at the baseline compared to the other groups. However, a previous study, which evaluated viral and host characteristics associated with reservoir size in 1057 individuals, found no association between initiation of ART beyond the first year after HIV infection and larger HIV reservoir size, independent of the time period of untreated HIV infection being analyzed [51], which applies to our study population. Our patients were long-term infected, and early ART was not a standard clinical practice at the time of their infection.

In addition, among the main design weaknesses, we highlight the small sample size being analyzed, in particular in HCV-exposed individuals, which could affect the statistical power. However, the massive administration of direct-acting antivirals in Spain from 2015 prevented us from collecting a higher number of samples from HIV/HCV-coinfected patients. Moreover, HCV positive samples have been limited for the past two years due to the health-system collapse caused by the SARS-CoV-2 pandemic, so more samples were unavailable at this time. Further studies including a higher number of patients will be of value in order to confirm our conclusions.

The use of qPCR to quantify the HIV reservoir could also be considered a shortcoming. However, although intra-individual variations of HIV DNA have been previously identified, when using qPCR instead of digital PCR [52], our results were carried out at least by quadruplicates in order to minimize that potential variation. Finally, the high cost of digital PCR does not allow it to be available, at least for the moment, in a large number of laboratories, so quantitative PCR is presented as the reference technique.

## 6. Conclusions

The elimination of HCV by DAAs seems unable to revert the consequences derived from chronic HCV infection, such as the increase in the reservoir size and viral splicing, which could indicate an increased risk of rapid HIV-reservoir reactivation. Therefore, the elimination of the HIV reservoir in HCV-infected PLWH might be hindered, and different approaches could be necessary.

## Figures and Tables

**Figure 1 jcm-11-03579-f001:**
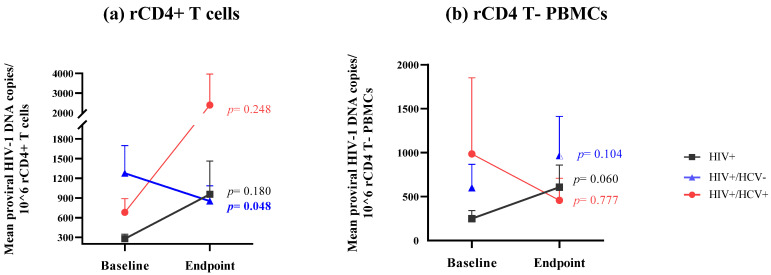
Evolution of HIV-reservoir size in (**a**) rCD4+ T cells and (**b**) rCD4 T- PBMCs. Footnote: The study groups were: (1) HIV+: PLWH; (2) HIV+/HCV-: PLWH and HCV, who spontaneously cleared HCV infection; and (3) HIV+/HCV+: PLWH and HCV, who were never treated for HCV at baseline but achieved a sustained viral response with direct-acting antivirals at endpoint. Symbols with connecting lines represent reservoir-size arithmetic mean and standard error of the mean. Differences between baseline and endpoint for the different study groups were calculated using a mixed-model Gamma-distributed GLM. Statistical significance was defined as *p* < 0.05 (2-tailed). HIV, human immunodeficiency virus; HCV, hepatitis C virus; rCD4+ T cells, resting CD4+ T cells; rCD4 T- PBMCs, resting CD4 T cells-depleted PBMCs; baseline, time of the study when HIV+/HCV+ individuals had never been treated for hepatitis; endpoint, time of the study when HIV+/HCV+ subjects had cleared HCV by treatment with direct-acting antivirals. HIV viral-splicing expression and dynamics.

**Figure 2 jcm-11-03579-f002:**
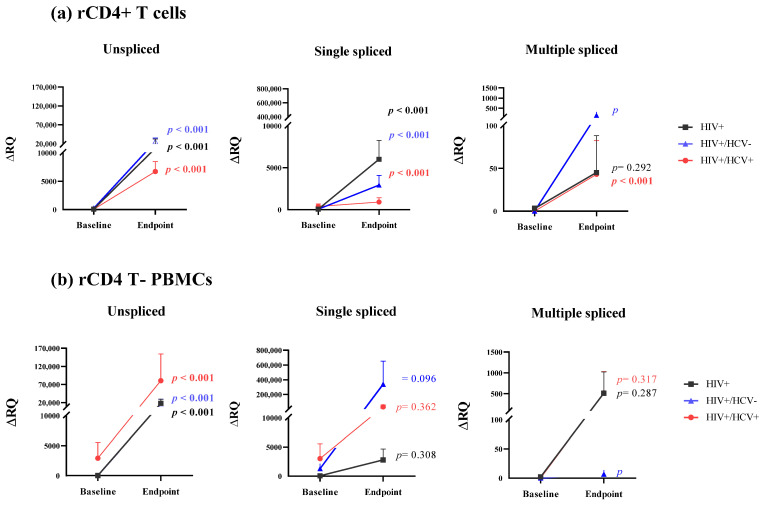
Evolution of HIV viral splicing in (**a**) rCD4+ T cells and (**b**) rCD4 T- PBMCs. Footnote: The study groups were: (1) HIV+: PLWH; (2) HIV+/HCV-: PLWH and HCV, who spontaneously cleared HCV infection; and (3) HIV+/HCV+: PLWH and HCV, who were never treated for HCV at baseline but achieved a sustained viral response with direct-acting antivirals at endpoint. Symbols with connecting lines represent reservoir-size arithmetic mean and standard error of the mean. Differences between baseline and endpoint for the different study groups were calculated using a mixed-model Gamma-distributed GLM. Statistical significance was defined as *p* < 0.05 (2-tailed). HIV, human immunodeficiency virus; HCV, hepatitis C virus; rCD4+ T cells, resting CD4+ T cells; rCD4 T- PBMCs, resting CD4 T cells-depleted PBMCs; baseline, time of the study when HIV+/HCV+ individuals had never been treated for hepatitis; endpoint, time of the study when HIV+/HCV+ subjects had cleared HCV by treatment with direct-acting antivirals.

**Table 1 jcm-11-03579-t001:** Epidemiological characteristics of the study population.

	TOTAL *n* = 63	HIV+ *n* = 22	HIV+/HCV- *n* = 17	HIV+/HCV+ *n* = 24	*p*
Male, *n* (%)	34 (54%)	13 (59.1%)	10 (58.8%)	11 (45.8%)	0.658
Age, years *	52 (44–55)	46 (37–54.3)	53 (49.5–55.5)	51.5 (44.3–55.8)	0.157
Weight, Kg *	65.9 (61.3–80.4)	66.8 (63.4–76.6)	66 (61–82)	63.5 (57.6–82.1)	0.632
Height, cm *	165 (160.172.5)	167 (160–174.5)	167 (162–175)	163 (160–169)	0.213
BMI, Kg/m^2^ *	24.7 (22–27.4)	25.6 (21.9–27.9)	24.8 (22–27.6)	23.9 (20.9–27.5)	0.854
Time of HIV infection, years *	20 (16.5–26.5)	12.1 (18–23.7) ^$^	21.1 (13.3–27.4)	24.1 (13.6–30)	** *0.035* **
Transmission route, *n* (%)					** *<0.001* **
IDUs	21 (33.3%)	0 (0%)	12 (70.6%)	9 (37.5%)	
MSM	11 (17.5%)	8 (36.4%)	2 (11.8%)	3 (12.5%)	
MSW	12 (19%)	8 (36.4%)	1 (5.9%)	3 (12.5%)	
Others	17 (27%)	6 (27.7%)	2 (11.8%)	9 (37.5%)	
CDC category, *n* (%)					0.325
A	32 (50.8%)	12 (54.5%)	7 (41.2%)	13 (54.2%)	
B	11 (17.5%)	3 (13.6%)	4 (23.5%)	4 (16.7%)	
C	14 (22.2%)	3 (13.6%)	4 (23.5%)	7 (29.2%)	
Unknown	6 (9.5%)	4 (18.2%)	2 (11.8%)	0 (0%)	
cART regimen, *n* (%)					0.409
NNRTIs	19 (30.2%)	7 (31.8%)	8 (47.1%)	4 (16.7%)	
NRTIs	2 (3.2%)	1 (4.5%)	0 (0%)	1 (4.2%)	
PIs	3 (4.8%)	1 (4.5%)	2 (11.8%)	0 (0%)	
INIs	27 (42.9%)	9 (40.9%)	4 (23.5%)	14 (58.3%)	
Dual therapy	5 (7.9%)	2 (9.1%)	1 (5.9%)	2 (8.3%)	
Monotherapy	1 (1.6%)	0 (0%)	0 (0%)	1 (4.2%)	
Unknown	6 (9.5%)	2 (9.1%)	2 (11.8%)	2 (8.3%)	
IFNL3 (IL-28B) genotype, *n* (%)					** *<0.001* **
CC	30 (47.6%)	9 (40.9%)	15 (88.2%)	6 (25%)	
Non-CC	26 (41.2%)	8 (36.4%)	2 (11.8%)	16 (66.7%)	
Unknown	7 (11.1%)	5 (22.7%)	0 (0%)	2 (8.3%)	

Footnote: HIV, human immunodeficiency virus type 1; HCV, hepatitis C virus; *n* (%), number (percentage); *, median (interquartile range); BMI, body mass index; IDUs, intravenous drug users; MSM, men who have sex with men; MSW, men who have sex with women/women who have sex with men; CDC, Center for Disease Control and prevention classification system for HIV infection; cART, combined antiretroviral therapy; NNRTIs, non-nucleoside reverse-transcriptase inhibitors; NRTIs, nucleoside analogue reverse-transcriptase inhibitors; PIs, protease inhibitors; INIs, integrase inhibitors; CMV, cytomegalovirus. For categorical variables, differences between study groups were analyzed via χ2 test (non-parametric). For continuous variables, normalization was studied with a Shapiro–Wilk test and differences between groups with an ANOVA test (parametric) and a Kruskal–Wallis H test (non-parametric). Statistical significance was defined as *p* < 0.05 (2-tailed). ^$^ One patient with missing time of HIV-infection data.

**Table 2 jcm-11-03579-t002:** Lymphocytic subpopulations of the study subjects.

	TOTAL *n* = 63		HIV+ *n* = 22	HIV+/HCV- *n* = 17	HIV+/HCV+ *n* = 24	*p*
CD4+ count, cells/µL *	B	783 (572–1099)	928 (698–1138)	752 (538–1026)	741 (558–1174)	0.398 ^$^
	E	835 (609–1117)	876 (700–1123)	808 (496–1018)	854 (602–1227)	
		**TOTAL** ***n* = 60**	**HIV+** ***n* = 20**	**HIV+/HCV-** ***n* = 17**	**HIV+/HCV+** ***n* = 23**	
CD4+, % *	B	36.5 (32.0–43.1)	39.5 (35.0–44.0)	36.0 (31.5–42.0)	33.0 (26.0–44.0)	0.308
	E	37.0 (31.0–42.4)	39.4 (34.3–44.3)	32.0 (25.5–39.5)	37.0 (27.4–42.3)	
	**TOTAL** ***n* = 14**		**HIV+** ***n* = 6**	**HIV+/HCV-** ***n* = 2**	**HIV+/HCV+** ***n* = 6**	
CD8+ count, cells/µL *	B	917 (803–1492)	904 (803–2492)	1028 (1028–1028)	991 (736–1849)	0.345 ^$^
	E	962 (794–1427)	904 (671–1341)	974 (974–974)	1175 (847–1844)	
	**TOTAL** ***n* = 14**		**HIV+** ***n* = 6**	**HIV+/HCV-** ***n* = 2**	**HIV+/HCV+** ***n* = 6**	
CD8+, % *	B	41.6 (35.9–46.9)	37.4 (34.7–43.2)	42.3 (42.3–42.3)	42.9 (40.1–54.8)	0.123 ^$^
	E	39.1 (33.9–45.4)	37.4 (32.8–41.6)	41.2 (41.2–41.2)	43.6 (38.9–53.4)	
	**TOTAL** ***n* = 14**		**HIV+** ***n* = 6**	**HIV+/HCV-** ***n* = 2**	**HIV+/HCV+** ***n* = 6**	
CD4:CD8 Ratio, *	B	0.93 (0.62–1.18)	1.05 (0.79–1.21)	0.88 (0.88–0.88)	0.77 (0.48–1.13)	0.735 ^$^
	E	0.96 (0.74–1.18)	1.05 (0.85–1.24)	0.86 (0.86–0.86)	0.88 (0.48–0.99)	

Footnote: HIV, human immunodeficiency virus type 1; HCV, hepatitis C virus; *, median (interquartile range); %, percentage; B, baseline; E, endpoint. At baseline, none of these patients had ever taken treatment for HCV infection. Normalization was studied with a Shapiro–Wilk test and differences between baseline and endpoint with a Student’s *t*-test for paired samples (parametric). ^$^ Wilcoxon test was used when deviation from a Gaussian distribution was observed. Statistical significance was defined as *p* < 0.05 (2-tailed).

**Table 3 jcm-11-03579-t003:** Clinical characteristics of the study population related to HCV infection.

	TOTAL *n* = 41	HIV+/HCV- *n* = 17	HIV+/HCV+ *n* = 24	*p*
HCV genotype				*n*.a.
GT1	12 (29.3%)		12 (50%)	
GT2	0 (0%)		0 (0%)	
GT3	2 (4.9%)		2 (8.3%)	
GT4	8 (19.5%)		8 (33.3%)	
Unknown/not applicable	19 (46.3%)		2 (8.3%)	
Fibrosis stage				0.066
F0–F1 (<6 kPa)	33 (80.5%)	12 (70.5%)	21 (87.5%)	
F2 (6–9 kPa)	1 (2.4%)	0 (0%)	1 (4.2%)	
F3 (>9–12 kPa)	1 (2.4%)	0 (0%)	1 (4.2%)	
F4 (>12 kPa)	0 (0%)	0 (0%)	0 (0%)	
Unknown/not applicable	6 (14.6%)	5 (29.5%)	1 (4.2%)	
HCV treatment				*n*.a.
Ledipasvir + Sofosbuvir	15 (36.6%)		15 (62.5%)	
Elbasvir + Grazoprevir	3 (7.3%)		3 (12.5%)	
Sofosbuvir + Daclatasvir	1 (2.4%)		1 (4.2%)	
Sofosbuvir + Velpatasvir	2 (4.9%)		2 (8.3%)	
Sofosbuvir + Simeprevir	1 (2.4%)		1 (4.2%)	
Viekirax + Exviera	1 (2.4%)		1 (4.2%)	
Unknown/not applicable	18 (43.9%)		1 (4.2%)	

Footnote: HIV, human immunodeficiency virus type 1; HCV, hepatitis C virus; *n* (%), number (percentage); kPa, kilopascals. **χ**2 test and the 2-sided Fisher exact test (non-parametric) were used to analyze differences between study groups. Statistical significance was defined as *p* < 0.05 (2-tailed).

## Data Availability

The data that support the findings of this study are available from the corresponding author (A.F.-R. or V.B.) upon reasonable request.

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
