# Peer review of "Dynamics of HIV Reservoir and HIV-1 Viral Splicing in HCV-Exposed Individuals after Elimination with DAAs or Spontaneous Clearance"

_jcm, 2022, doi:10.3390/jcm11133579_

Round 1

Reviewer 1 Report

Ms. Gianna Gao,

Assistant Editor of Journal of Clinical Medicine

Dear Ms. Gao,

The Manuscript entitled “Dynamics of HIV reservoir and HIV-1 viral splicing in HCV exposed individuals after elimination with DAAs or spontaneous clearance” showed important changes in HIV reservoir size in rCD4+ T cells in the three groups of PLWH, especially the reduction of the HIV reservoir size in HIV+/HCV- spontaneous clarifiers between baseline and the final endpoint (after 48 weeks). In addition, when evaluating multiple-spliced transcripts, coinfected subjects showed a significant increase in the mean expression, in contrast with spontaneous clarifiers and HIV-monoinfected individuals. This results are very important for the comprehension of the HIV/HCV coinfection, especially for countries with a high burden of both virus. It is important to highlight the difficulty in carrying out and organize such a study, especially to include coinfected subjects naïve of treatment. I really enjoyed reading this manuscript. It is well written, scientifically sound, and I couldn’t find any major flaw in the methods. Find below few suggestions in order to improve the manuscript before publication.

Specific comments:

1- Line 64, Page 2: Please, describe the full name of abbreviations (PLWH) as they first appears in the text.

2- Line 130, Page 3: Please, delete the duplicate words “as previously described”.

Reviewer 2 Report

Congratulations to the authors for their outstanding work.
Study planning could have been done better, as could the comparison of groups and results.
Discussions should be improved by showing all design weaknesses, how groups and results are compared, and the limitations imposed by statistics.
Overall, the study is interesting. A correction of the article by an English speaker is required.

Reviewer 3 Report

This manuscript addresses the interesting topic of the changes in HIV reservoir among patients with HIV/HCV co-infection. The strength of the study is that it included three arms: HIV+ mono-infected, HIV/HCV spontaneous clearance, HIV/HCV treated with DAA. 

Although the paper was written well, one of the limitations of the study is the relatively low sample size. 

My questions/comments are as follows:

1. Some publications have implied that the HIV viral reservoir may be impacted by INSTI, due to its inherent mechanism of action of blocking host genome viral integration. Although Table 1 indicates that there were no statistically significant difference in antiretroviral regimen between the groups, I suggest to check differences in HIV reservoir and HIV viral splicing between patients on INSTI vs non-INSTI regimen. 

2. The HIV reservoir size for paired samples were expressed in mean and standard deviation. The standard deviations are wide. 

  2a. A statistically significant decrease in HIV reservoir size was found in HIV/HCV- spontaneous clarifiers. However, the SD was wide and the p-value almost straddled non-significance (p=0.048). The Methods indicated that paired t-test was used. Was the data on HIV reservoir size normally distributed? If not, non-parametric test is more appropriate. Would the result still be significant if alternative analyses are used?

b. Given the wide standard errors, I suggest to please include a table of to show each patient's pro-viral DNA copies at baseline vs endpoint. This way, the readers would be able to see whether there are outliers that could be affecting the results.

3. A study on HIV dynamics on HIV/HCV co-infected patients in Spain concluded that "HIV‑reservoir size is not afected either by HCV coinfection or by direct acting antivirals (DAAs) therapy" (Beatriz Alvarez, et al, Scientific Reports, 2022). Please include this paper in the discussion and compare/contrast their results with the current study.   

4. In the Limitations paragraph, 'Study group 1 shows shorter period of HIV infection.... analyses were corrected by this factor." Please clarify/reference which part of the Methods section showed adjustment of the analyses using the variable HIV duration.  

Minor comments include:

> Suggest to remove brand names (Harvoni, etc) of the HCV treatment regimens in Table 3.

>  Under 'HCV Treatment', Column 3 (HIV+/HCV-) should include '17 (100%)' so that column total of '18' would add up.

> Include the small sample size in the Limitations section. 
